# How to Make Digitalization Better Serve an Increasing Quality of Life?

Oleg A. Kryzhanovskij [1], Natalia A. Baburina [2] and Anastasia O. Ljovkina [1,3,*]

1   Department of Economic Security, System Analysis and Control, Financial-Economic Institute,
    University of Tyumen, 625007 Tyumen, Russia; o.a.kryzhanovskij@utmn.ru
2   Department of Economics and Finance, Financial-Economic Institute, University of Tyumen, 625007 Tyumen,
    Russia; n.a.baburina@utmn.ru
3   Institute of Advanced Manufacturing Technologies, Peter the Great St. Petersburg Polytechnic University,
    195251 Saint Petersburg, Russia
*   Correspondence: a.o.lyovkina@utmn.ru or anastasia.ljovkina@gmail.com; Tel.: +7-982-903-8395

**Abstract:** Modern people live in the era of knowledge and digitalization supposed to increase their quality of life. Nevertheless, digital technologies are only the instruments in the development and transformation of social-economic processes and their usage per se does not ensure only positive effects, which much depends on goals, conditions, institutes, etc. Thus, digitalization has an unambiguous influence on many social-economic processes and needs a wise policy to provide smooth progress and well-being for everybody. This study aims to design and test appropriate tools for managing digitalization to direct this process on increasing the quality of life. For this purpose we analyzed: (1) correlation to identify interrelations between digitalization and quality of life; (2) the potential of using the visualization matrix method to identify and monitor national trends of digitalization in the context of quality of life. We found: (1) close correlation between subjective and objective indicators of quality of life and between the quality of life and digitalization; (2) the two-dimensional matrix turned out to be a relevant visual tool that embraces specific two-way relationships between human development and digitalization. In combination with statistical and qualitative methods, this tool has wide prospects for managing digitalization in the context of social progress and increasing quality of life.

**Keywords:** quality of life; digitalization; human development; ranking of happiness; HDI; world digital competitiveness ranking

## 1. Introduction

Digitalization being a new level of technological progress and a new step in the knowledge era is supposed to be aimed at increasing the quality of life. Digital technologies have already passed from the status of being just an instrument to the status of a modifying factor changing social life in its every sphere. External digitalization expression is an expansion of the use of information and communication technologies and the implementation of digital technologies in human life, which act as a catalyst for the information society development. Digitalization not only transforms the way people interact with the world around them, but also changes their internal world: attitude to the world, to oneself, and to what it means to be human [1]. Information and communication technologies have an ever-increasing influence on human self-esteem, mutual actions, and socialization, the concept of reality and its perception, as well as interaction with reality [2]. Thus, in the modern world, digitalization acts as one of the key determinants of human development, affecting not only objective indicators of the quality of life but also its subjective perception.

Digital technologies have a great potential for increasing the quality of life as they can sufficiently extend human development opportunities [3–8]. In the economy of the 21st century driven by innovations, services, and intangible goods, this impact implies not

only economic growth and welfare measured by GDP [9], but also increased value of time which less and less spent on routine, opportunities for creativeness and self-development, and the emergence of new, often "free" goods as a result of the digital revolution [10]. The use of digital technologies contributes to the improvement of the well-being of the country, the development of social capital and the achievement of social equality, providing access to health-related information, medical services, education for the poor, and facilitates trade [11].

Quality of life is determined by the ability to build up social capital, achieve professional goals, receive a quality education, and develop interpersonal relationships and connections. Existing research on the interrelations of digitalization and quality of life argues that increased access to information and communication technologies has a positive impact on the quality of life [12]. But is this influence unambiguous? Digital innovation undoubtedly impacts human life by saving time, spreading knowledge, increasing the availability of communication, enhancing network interactions, and automation through implementing artificial intelligence and big data technologies, increasing productivity and access to information, reducing deprivation, improving transparency and governance, creating social capital, and empowering people. Together with that, the positive effects of digitalization processes go together with social, economic, and psychological threats to the individual, society, state, and the world community. They increase risks of cyber threats and insecurity of privacy, unemployment, digital inequality, and encourage sedentary lifestyles [13]. While the digital economy drives productivity gains and positively impacts local and global economies, digitalization raises potential sustainability problems related to social and environmental well-being, driven by the automation of information processing and service delivery [14]. Automation and the widespread adoption of new technologies exacerbate the destruction of traditional spheres of activity, structural changes in the economy and, as a consequence, unemployment and inequality in wages, which bring about well-being problems [15–17]. An analysis of cases in the industrial area shows that even small efforts in the digitalization of processes can create problems that affect well-being and productivity at different stages of the digital technologies' implementation [18]. The emergence of a new digital segment of the shadow economy multiplies theft of personal data, financial resources, databases, and other cyber risks [19,20].

Many researchers point out that the positive or negative influence of digitalization on the quality of life sufficiently depends on the social-economic and political conditions and frames. In particular, digitalization gives rise to the problem of the digital divide and digital inequality, manifested in unequal access to information and communication technologies for different social strata of the population, countries, and peoples of different levels of economic development. Whether information technology is considered a public or private good depends in part on the civic culture of the community [21], the type of political regime and the extent to which democracy has spread [22], and existing social and economic inequalities [23]. The digital divide is mainly explained by income differentials [24], scientific and technological potential [25] and exacerbated by differences in IT skills and the degree of technical savvy [26,27]. Furthermore, the impact of digitalization on well-being is increasingly dependent on how we use digital tools, and in some cases, on our limitations on their use [28]. In recent research, it was marked that digitalization does not turn the state into a country of prosperity [29]. It contributes to increased national wealth only if the country has an adequate education system, good governance, and a philanthropic financial system [30].

In addition, there are categories of citizens vulnerable to rapid technological changes in society. In particular, older people in their daily lives face the accelerated pace of the digitalization of services, which increases their feelings of anxiety and undermines their well-being [31]. The transition of many social processes to online forms exposes the problem of the lack of skills of older people for effective participation in the digital environment, which is often hampered by limited physical mobility and a decrease in their social networks and contacts [32]. The digital divide limits the ability to use IT to form

social bonds that positively affect human well-being. In particular, the COVID-19 pandemic showed that older people, people without Internet access, and people with limited Internet access skills are staying away from using digital communications at a time when the use of such communication is especially important [33].

The explosive growth of the internet and social media has an ambivalent impact on subjective well-being [34–37]. According to A. Clark, the vector of the influence of social networks on subjective well-being is determined by the structure of interpersonal connections and behavior [35]. The usage of social media to form and expand meaningful social connections has a positive effect on the well-being of users, as it responds to the innate human desire for acceptance and belonging. Nonetheless, if user behavior is not aimed at establishing social connections, then social networks can negatively affect the well-being of users, through such traps as isolation and social comparison [35].

Thus, digitalization has created new challenges for the development of society, questioning the perspective of human quality of life. Negative aspects of the impact of digitalization on quality of life are leveled out in societies that attach great importance to education and training, culture, civic activities, health, and equal development opportunities. Based on the general logic, we can assume that digitalization does not increase the quality of life and contribute to low-quality human development by itself, being a neutral process of the widespread introduction of digital technologies in essence. Thus, it cannot act as an exhaustive, sufficient determinant of improving the quality of life and should be analyzed only in a combination with certain goals, social-economic and political conditions, and wise management. The management of the quality of life implies taking into account, first of all, fundamental factors of social policy. However, the focus of this study is digitalization as an actual modern social context that allows expanding the possibilities for improving the quality of life in all spheres of human life, and as an environment that forms new realities and a qualitatively new potential for applying management decisions for the benefit of people.

This study aims to design the appropriate tools for managing digitalization to direct this process on increasing quality of life. For this purpose, we analyzed: (1) the correlation to identify interrelations between digitalization and quality of life; (2) the potential of using the visualization matrix method to identify and monitor national trends of digitalization in the context of quality of life.

## 2. Materials and Methods

### 2.1. Indicators

The existing pluralism of the approaches to assessing the quality of life is both due to the ambiguity of the concept itself and the complex, multidimensional character of this category. Thus, existing approaches to assessing the quality of life are multi-criteria and include a range of indicators, selected in accordance with the research hypothesis of the authors. There are many perspective studies on the development of a synthetic measure of the quality of life [38–45].

In this study, we chose two popular indicators that reflect the quality of life from both a subjective and objective standpoint. We used a ranking of happiness (RH) to investigate the correlation between subjective well-being and objective conditions for human development (HDI). The World Happiness Report is a publication of the Sustainable Development Solutions Network, powered by data from the Gallup World Poll [46]. In addition, we used the human development index (HDI) as a measure of the quality of life and world digital competitiveness ranking (WDCR) as a measure for digitalization.

Since the 1990s, with issuing the first HDR, welfare growth has ceased to be considered the only and the main indicator of the quality of life. Rather, the growth of well-being has moved to the category of the means of development. The quality of life has come to be seen through the prism of human development. The HDR methodology put humans at the head of social development, thereby arguing that social development should be determined not only by economic growth but also by providing more opportunities and freedoms for life.

The development process itself must create the conditions for the opportunity to reveal and realize human potential individually or collectively and for a productive and creative life.

The HDR includes the human development index (HDI) as the main complex indicator for assessing the quality of life. HDI is based on the key foundations of human development: a long and healthy life, knowledge, and access to resources to ensure a decent standard of living. Furthermore, it considers the problem of the equality of all people in their rights to a high quality of life. The 2019 HDR press release talks about shifting the focus on inequality, from not only inequality in income, but also to inequalities in other dimensions, such as health, education, access to technology, and exposure to economic and climate shocks.

HDR methodology was widely adopted at the national and regional levels. The Analytical Center under the Government of the Russian Federation has been monitoring HDI by regions since 2008 (UNDP 2010). The calculation of the index for Russian regions is carried out according to the UNDP methodology considering the availability of data necessary for calculating HDI and specifics of regional statistics [47].

There are different quantitative tools useful for managing and monitoring national digital development. The first category of these tools is composite indices allowing the capture and monitoring of national digital readiness and competitiveness and comparing digitalization degrees across countries. For example, the Digital Government Index (DGI) prepared by OECD measures the maturity level of digital government strategies in OECD member and partner countries based on evidence gathered through the Survey on Digital Government. The Digital Economy Index (DEI) measures inflation in what people are buying in the digital world in major global economies. The Cisco Digital Readiness Index was developed to measure a country's level of digital readiness by seven holistic components: basic needs, business, and government investment, ease of doing business, human capital, start-up environment, technology adoption, and technology infrastructure. The Digital Economy and Society Index (DESI) is a composite index reflecting Europe's digital performance and competitiveness. DiGiX and World Digital Competitiveness Ranking (WDCR) are used globally to capture national digitalization status. DiGiX includes 19 indicators grouped in 6 dimensions that represent 3 broad pillars: supply conditions, demand conditions, and institutional environment.

In our research, we used WDCR, which measures the capacity and readiness of a national economy to adopt and explore digital technologies as a key driver for economic transformation in business, government, and wider society. The rankings are calculated based on the 50 ranking criteria: 30 Hard and 20 Survey data. The rating is assigned according to the cumulative result, shown in 3 categories: (1) "Knowledge"—the quality of education and science, (2) "Technologies"—development of Internet and communication technologies, financial capital in the IT industry, as well as the regulatory environment, (3) "Future Readiness"—the level of readiness to use digital transformation [48].

The second big category of the tools useful for managing and monitoring national digital development are statistic and mathematical multi-factor models capturing interrelations of factors including digitalization and technological progress and using digitalization indicators and composite indices. The big variety of these models is stipulated by their different scales, purposes, context, included factors, and used data. For example, Ronald Paul Hill and Kanwalroop Kathy Dhanda used the Technological Achievement Index (TAI) and HDI to examine the relationship between technological achievement and human development so that the human rights community may better understand the impact of the digital divide worldwide. ANOVA analysis revealed the strength of the interdependence between them, especially among the least developed nations [49]. A dynamic panel study on digitalization and a firm's agility allowed to better understand factors driving agility in advanced economies, including at national/industry digitalization level [50]. The effects of digital transformation on value creation were studied based on indicators: technology readiness (e.g., ICT investments), digital technology exploration (e.g., research and development), and digital technology exploitation (e.g., patents and trademarks). The research

identified several significant relationships between such constructs, which contribute to the literature and provide key implications for business management and practitioners [51].

Nevertheless, composite indices and statistical analysis cannot provide comprehensive information for the effective management of digitalization processes directing them on increasing the quality of life. On the other hand, qualitative analysis needs insights in identifying areas of possible problem issues and solutions and purpose quantitative reference points. Thus, to our mind, there is a lack of intermediate tools using both quantitative approaches and visual analysis and considering unambiguous interrelations between digitalization and quality of life, which allows to identify possible problem areas in national digital transformation, and clarify directions for further qualitative studies and developing quantitative models.

### 2.2. Research Procedure and Methods

The sample size was 61 countries that had RH, HDI, and WDCR indices. Like other digitalization indexes, WDCR is a relatively recent index, thus we made the analysis on the example of the most recent period, in which there were all required reports with RH, HDI, and WDCR indices—in 2018.

As the first step, we analyzed the relationships between subjective well-being and objective conditions for human development with the help of standard Pearson correlation coefficient and visual matrix analysis. We used a two-dimensional matrix (RH/HDI) divided into four quadrants by medians. The median HDI along the abscissa and the median RH along the ordinate axis were calculated from all relevant observations for the period 2018.

As the second step, we analyzed the relationships between the HDI and WDCR using correlation analysis and a two-dimensional matrix (WDCR/HDI) divided into four quadrants by medians. The median WDCR along the abscissa and the median HDI along the ordinate axis were calculated from all relevant observations for the period 2018.

As the third step, we grouped countries into four groups according to the four quadrants of the WDCR/HDI matrix.

In our research, we used open-source empirical data: HDI report [47], World Digital Competitiveness Ranking [48], and the World Happiness Report [46]. We made all statistical calculations and matrix visualization using open-source software R, the PLM package (GNU license).

## 3. Results

### 3.1. The Results of Correlation and Visual Matrix Analysis

Correlation analysis showed a close positive linear correlation between subjective well-being and improving conditions for human development.

The coefficient of determination ($R^2$) 64% in 2018 indicates rather strong positive relationships between the objective conditions and opportunities for human development and subjective well-being and happiness, which allows choosing any of these indicators for further investigation of the relationships between digitalization and quality of life (Figure 1). Nevertheless, several countries lie far from the common correlation line.

The coefficient of determination ($R^2$) 75% indicates a rather strong positive correlation between world digital competitiveness ranking (WDCR) and HDI (Figure 2).

The coefficient of determination ($R^2$) 49% indicates a moderate positive correlation between WDCR and ranking of happiness (RH) (Figure 3).

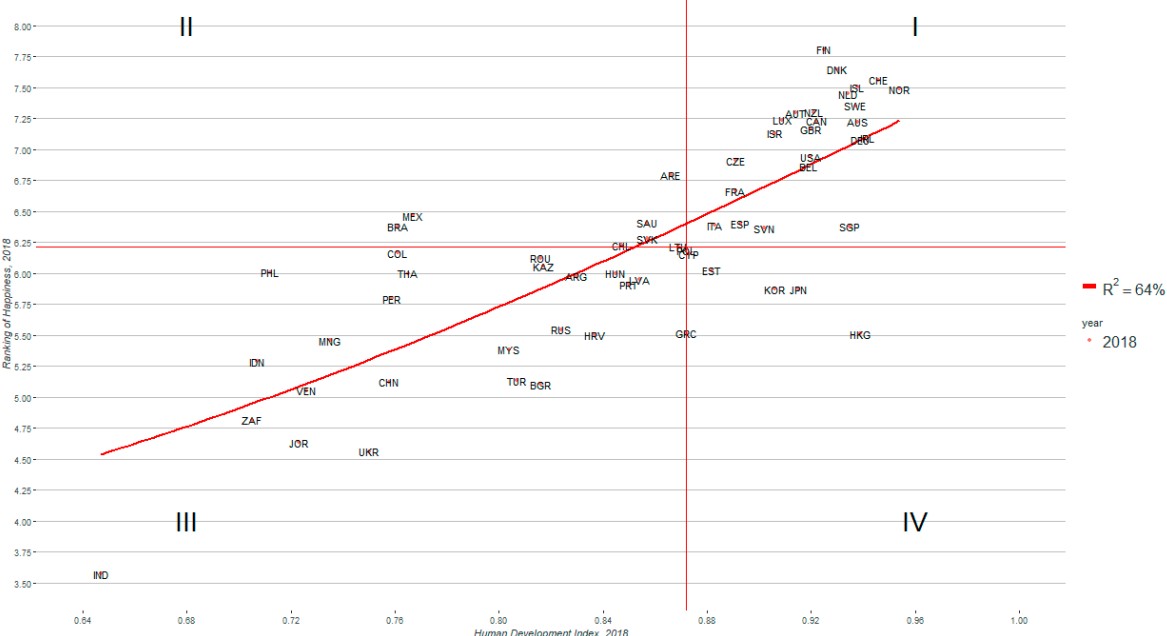

**Figure 1.** Correlation between RH and human development index (HDI) at the two-dimensional matrix, 2018. Note: country codes are based on the standard ISO 3166 [52].

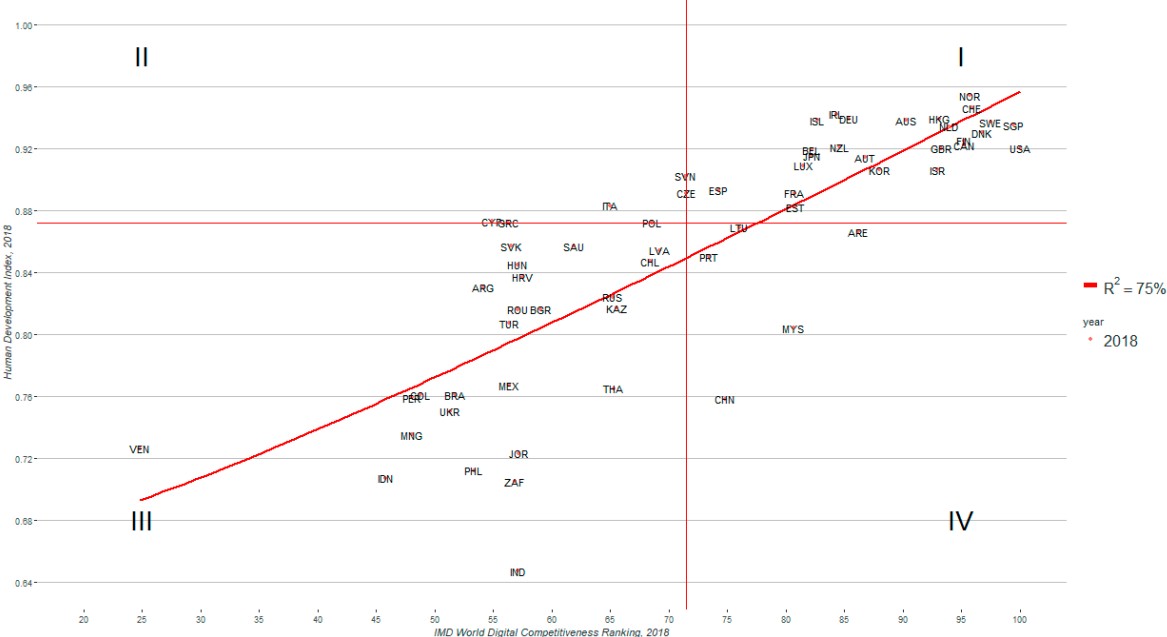

**Figure 2.** Correlation between HDI and WDCR at the two-dimensional matrix, 2018. Note: country codes are based on the standard ISO 3166 [52].

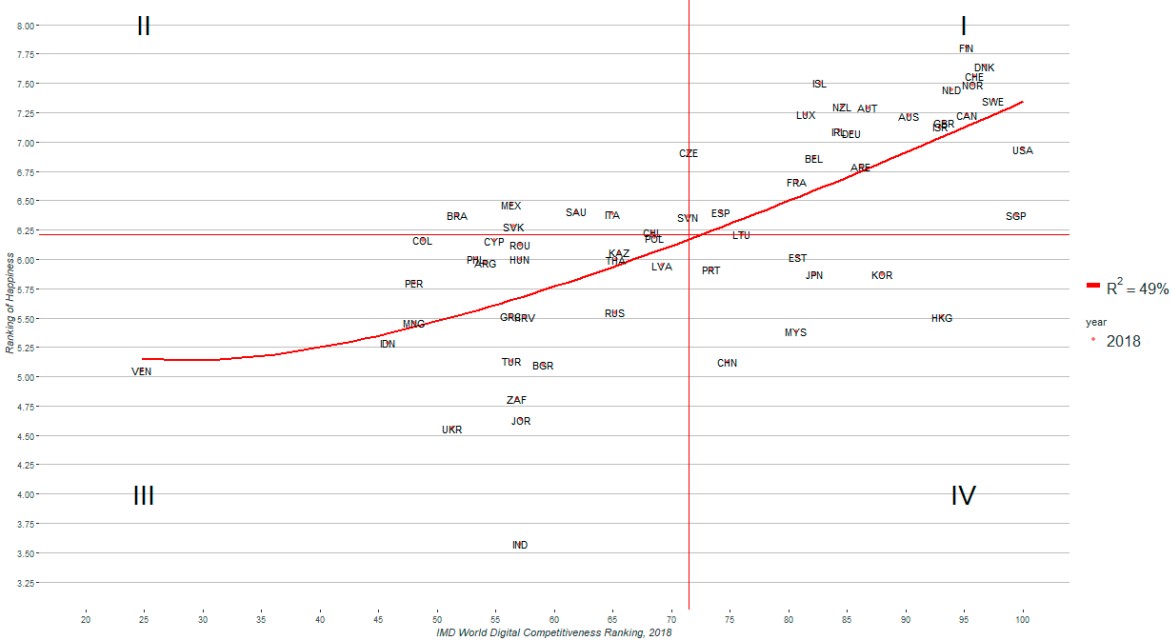

**Figure 3.** Correlation between RH and WDCR at the two-dimensional matrix, 2018. Note: country codes are based on the standard ISO 3166 [52].

### 3.2. Classification of Countries by the Level of HDI and WDCR

We classified countries into four conditional groups by the level of HDI and WDCR (Table 1).

**Table 1.** Classification of countries by the level of HDI and WDCR.

| WDCR | HDI | |
| --- | --- | --- |
| | **Reduced** | **Increased** |
| Increased | Group B (4): CZE, ITA, SVN, *CYP* | Group A (25): AUS, AUT, BEL, CAN, CHE, DEU, DNK, ESP, FIN, FRA, GBR, IRL, ISL, ISR, LUX, NLD, NOR, NZL, SGP, SWE, USA, *EST, HKG, JPN, KOR* |
| Reduced | Group C (27): *BRA, CHL, MEX, SAU, SVK,* ARG, BGR, COL, GRC, HRV, HUN, IDN, IND, JOR, KAZ, LVA, MNG, PER, PHL, POL, ROU, RUS, THA, TUR, UKR, VEN, ZAF. | Group D (5): *ARE,* CHN, LTU, MYS, PRT. |

Note: Country codes are based on the standard ISO 3166 [52]; countries are included in different groups in Tables 1 and 2 in italics.

**Table 2.** Classification of countries by the level of RH and WDCR.

| WDCR | RH | |
| --- | --- | --- |
| | **Reduced** | **Increased** |
| Increased | Group B (8): CZE, ITA, SVN, *BRA, CHL, MEX, SAU, SVK* | Group A (22): AUS, AUT, BEL, CAN, CHE, DEU, DNK, ESP, FIN, FRA, GBR, IRL, ISL, ISR, LUX, NLD, NOR, NZL, SGP, SWE, USA, *ARE* |
| Reduced | Group C (23): *CYP,* ARG, BGR, COL, GRC, HRV, HUN, IDN, IND, JOR, KAZ, LVA, MNG, PER, PHL, POL, ROU, RUS, THA, TUR, UKR, VEN, ZAF | Group D (8): *EST, HKG, JPN, KOR,* CHN, LTU, MYS, PRT |

Note: country codes are based on the standard ISO 3166 [52]; countries are included in different groups in Tables 1 and 2 in italics.

In the WDCR/HDI matrix, groups A and B are the most numerous. Group A includes 25 countries with increased HDI and increased WDCR and Group B includes 27 countries with increased HDI and reduced WDCR. Only four countries have reduced HDI whereas WDCR is higher than average. Group D consists of five countries with increased HDI and reduced WDCR.

In the WDCR/RH matrix, groups A and B are also the most numerous but contain fewer countries than WDCR/HDI matrix. Group A includes 22 countries with increased RH and increased WDCR and Group B includes 23 countries with increased RH and reduced WDCR. Only eight countries have reduced RH whereas WDCR is higher than average. Group D consists of eight countries with increased RH and reduced WDCR.

## 4. Discussion

### 4.1. Results Discussion

The correlation analysis confirms modern ideas about the close relationship between digitalization and quality of life [3–8,11]. At the same time, visual matrix analysis confirmed that a high level of digitalization does not obligatorily ensure a high level of quality of life [29] as there were a sufficient amount of countries with increased WDCR and reduced HDI and RH. Furthermore, several countries had increased HDI or RH at reduced WDCR, vice versa. These cases prove that digitalization is undoubtedly a significant process in changing the quality of life, but it needs to be taken into account together with the policy and conditions of digitalization and other sufficient social-economic factors, which are more fundamental than digitalization itself [21–25].

With the help of two-dimensional matrixes, we mapped and classified countries into four conditional groups, which allowed us to see an overall comparative picture of countries' distribution by the level of digitalization in the context of quality of life. Thus, the matrix can be used in developing digitalization policy as a relative map of the countries' routes of digitalization in the context of quality of life and a field of countries' comparison. For example, two or several countries with the same HDI and different levels of WDCR can be compared for finding the reserves and best practices to improve, implying digitalization for increasing quality of life. In addition, this comparison and additional qualitative analysis allows identifying sufficient factors that influence quality and factors that allow directing digitalization on increasing HDI and RH most effectively.

Thus, the statistical approach alone is not relevant for managing the digitalization process for increasing quality of life. Alternatively, we proposed the management tool based on a non-econometric approach. WDCR/HDI and WDCR/RH matrixes visualize national digitalization in a coordinate system of human development and a subjective feeling of happiness, which allowed comparing the country social progress (in terms of quality of life) in the digital economy. We concluded that this tool is rather useful for managing and monitoring national digital development. It allows mapping different countries in the same coordinate system, making typological and comparative analysis. However, it should be noted that the usage of this tool is limited by the need to supplement it with qualitative analysis for considering specific national factors.

Thus, in this study, we tried statistical and visual approaches to analyze digitalization in the context of quality of life for suggesting relevant instruments. Correlation analysis showed a close relationship between digitalization and quality of life and it can be considered as a sufficient factor in further panel data models explaining the national quality of life. Nonetheless, we discovered more potential in the use of visualization tools that allow graphically mapping countries positions in the dimensions of digitalization and quality of life. In particular, the suggested two-dimensional matrixes allow identifying the field for additional qualitative analysis in country comparison and allow to explore the factors sufficient for making digitalization better serve increasing the quality of life. We recommend this tool for developing national digital economy policies and programs serving better social progress.

*4.2. Research Limitations and Direction of Further Research Development*

The research uses data only from one period, the year 2018. The dynamic analysis in further research and developing panel models can show the greater potential of statistical instruments in managing the processes of digitalization in the context of quality of life. Futhermore, dynamic analysis will allow us to visualize national digital development trajectories and trends in the context of quality of life.

The research sample is limited by the chosen indicators. Nevertheless, we used two alternative indicators of subjective well-being and objective conditions for human development to show the potential of the used instrument. In further research, the influence of various components of the WDCR can be investigated.

The next limitation of the study is the list of suggested measures, which can be extended and added due to additional qualitative and quantitative studies. In particular, we plan to develop a case study on the example of Russian digitalization policy in our further research.

Furthermore, the research results do not clarify how to implement the suggested measures and do not provide certain decision-making technology in the digitalization policy, which can be a subject of subsequent research and practice.

The indices used in this study complement the standard statistical macro and micro indicators: per capita income, consumption, the share of consumption in income, leisure, inequality, and several others [53]. Nevertheless, new contexts reveal new dialectical limitations and require further development of both concepts of the quality of life and approach for the assessment of the quality of life, digitalization, and technological progress. Thus, in the context of the coronavirus pandemic, completely new indicators of the quality of life are coming to the fore: the availability of tests and vaccines for COVID-19, the availability of fast internet for permanent work in a remote office, the prevalence and quality of delivery services, and many others. According to C. Graham, pre-existing inequalities have exacerbated in the United States since the start of the pandemic, which is reflected in a deep decline in well-being. It was concluded that economic growth alone is not enough to support the economy and society [54]. The digitalization of everyday life is reaching a qualitatively new level and requires measurement by new indicators. Attempts to use old measures to take into account the levels of happiness, human and digital development in the new post-COVID conditions of life and work can lead to wrong political decisions and, as a result, ineffective spending of the limited financial funds and other resources. Thus, the overall limitation of the research aimed at designing the tools for the management of digitalization processes and quality of life is the growing incompatibility of standard economic data before and after 2020, continuous developing of the concept of the quality of life and digitalization and tools for its assessment.

This article does not claim to be an exhaustive study of the impact of digitalization on the quality of life. The proposed tool allows identifying the existing relationships between the phenomena under consideration and to determine the area for further qualitative and quantitative research that can be used to monitor the concordance of technological and social policies and form the basis for making management decisions to improve the quality of life.

**Author Contributions:** A.O.L. and N.A.B. developed the concept and design of the research, made the theoretical review, contributed to the analysis and discussion of the results. O.A.K. processed data and contributed to the result analysis, discussion, and research limitations. All authors have read and agreed to the published version of the manuscript.

**Funding:** The research is partially funded by the Ministry of Science and Higher Education of the Russian Federation as part of World-class Research Center program: Advanced Digital Technologies (contract No. 075-15-2020-934 dated 17.11.2020).

**Conflicts of Interest:** The authors declare no conflict of interest.

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
