# Peer review of "How to Make Digitalization Better Serve an Increasing Quality of Life?"

_sustainability, doi:10.3390/su13020611_

Round 1
Reviewer 1 Report
The method is impressive, the authors use original approach to the quality of life measurement
I appreciate that the paper suggests to use correlation to identify and monitor the unambiguous interrelations between digitalization and quality of life which hasn’t been done before in many research papers. The authors visualize matrix method to identify and monitor national trends of digitalization in the context of quality of life.
I would recommend to elaborate the policy and conditions of digitalization and other sufficient social-economic factors, which are more fundamental for the countries’ social policy than digitalization itself.
The authors make a very important conclusion that statistical approach alone is not relevant for managing the digitalization process for increasing quality of life. However, the main official approach for the quality of life measurement used in many countries to date is statistical. In this regard, it would be very interesting to know the alternative methodologies of measuring the quality of life and the indicators the authors suggest.
I would recommend to describe the tools which is considered useful for managing and monitoring national digital development in more details.
Author Response
Answers to the reviewer in the uploaded file.

Reviewer 2 Report
This is an interesting work, based on existing studies. It brings new insights by studying correlations between indices, but in my opinion it should also present at least one case study (e.g., in the author's own country) to improve the quality of its conclusions.
Some other suggestions include: replacing "digitization" with "digitalization" in the two instances it appears, and discussing in the limitations section about the possible third variables that might influence both sides of the correlations. Also, I would recommend changing "high" and "low" in the tables, figures and text discussing the correlations (maybe with "reduced" and "increased"), since it is confusing for properly reading the HDI values, which in itself uses these terms.
Author Response

(The authors gave the same response as above.)
